# LOOPED TRANSFORMERS ARE BETTER AT LEARNING LEARNING ALGORITHMS

**Liu Yang, Kangwook Lee, Robert D. Nowak & Dimitris Papailiopoulos**
University of Wisconsin, Madison, USA
`{liu.yang, kangwook.lee, rdnowak}@wisc.edu, dimitris@papail.io`

## ABSTRACT

Transformers have demonstrated effectiveness in *in-context solving* data-fitting problems from various (latent) models, as reported by Garg et al. (2022). However, the absence of an inherent iterative structure in the transformer architecture presents a challenge in emulating the iterative algorithms, which are commonly employed in traditional machine learning methods. To address this, we propose the utilization of *looped* transformer architecture and its associated training methodology, with the aim of incorporating iterative characteristics into the transformer architectures. Experimental results suggest that the looped transformer achieves performance comparable to the standard transformer in solving various data-fitting problems, while utilizing less than 10% of the parameter count.[1]

## 1 INTRODUCTION

Transformers (Vaswani et al., 2017; Brown et al., 2020; Devlin et al., 2019) have emerged as the preferred model in the field of natural language processing (NLP) and other domains requiring sequence-to-sequence modeling. Besides their state-of-art performance in natural language processing tasks, large language models (LLM) such as GPT-3 (Brown et al., 2020) and PaLM (Chowdhery et al., 2022) also exhibit the ability to learn in-context: they can adapt to various downstream tasks based on a brief prompt, thus bypassing the need for additional model fine-tuning. This intriguing ability of in-context learning has sparked interest in the research community, leading numerous studies (Min et al., 2022; Olsson et al., 2022; Li et al., 2023). However, the underlying mechanisms enabling these transformers to perform in-context learning remain unclear.

In an effort to understand the in-context learning behavior of LLMs, Garg et al. (2022) investigated the performance of transformers, when trained from scratch, in solving specific function class learning problems in-context. Notably, transformers exhibited strong performance across all tasks, matching or even surpassing traditional solvers. Building on this, Akyürek et al. (2022) explored the transformer-based model's capability to address the linear regression learning problem, interpreting it as an implicit form of established learning algorithms. Their study included both theoretical and empirical perspectives to understand how transformers learn these functions. Subsequently, von Oswald et al. (2022) demonstrated empirically that, when trained to predict the linear function output, a linear self-attention-only transformer inherently learns to perform a single step of gradient descent to solve the linear regression task in-context. While the approach and foundational theory presented by von Oswald et al. (2022) are promising, there exists a significant gap between the simplified architecture they examined and the standard decoder transformer used in practice. The challenge of training a standard decoder transformer from scratch, with only minor architectural modifications, to effectively replicate the learning algorithm remains an open question.

In traditional machine learning, *iterative* algorithms are commonly used to solve linear regression. However, the methodologies employed by standard transformers are not naturally structured for iterative computation. A *looped* transformer architecture, extensively studied in the literature such as Giannou et al. (2023), provides a promising avenue to bridge this gap. In addition to its inherent advantage of addressing problem-solving in an iterative manner, the looped transformer also breaks down tasks into simpler subtasks, potentially leading to significant savings in model parameters.

To illustrate how task breakdown leads to saving parameters, let's look closely at using the transformer model to solve linear regression task, specifically $\boldsymbol{w}$ in $\min_{\boldsymbol{w}} \|\boldsymbol{X}\boldsymbol{w} - \boldsymbol{y}\|_2^2$ (Fig. 1). To train a transformer on this task, we input a prompt sequence formatted as $(\boldsymbol{x}_1, \boldsymbol{w}^T\boldsymbol{x}_1, \ldots, \boldsymbol{x}_k, \boldsymbol{w}^T\boldsymbol{x}_k, \boldsymbol{x}_{\text{test}})$. Here $\boldsymbol{w}$ represents the parameters of the linear regression model, $\{\boldsymbol{x}_1, \cdots, \boldsymbol{x}_k\}$ are $k$ in-context samples, and $\boldsymbol{x}_{\text{test}}$ is the test sample. The transformer can potentially try to predict $y_{\text{test}}$ by approximating the ordinary least squares solution in a single forward pass. The computation of the matrix inverse, as required in the ordinary least squares solution $(\boldsymbol{X}^T\boldsymbol{X})^{-1}\boldsymbol{X}^T\boldsymbol{y}$, is more difficult for transformers to

---

[1]Our code is available at `https://github.com/Leiay/looped_transformer`.

learn compared to matrix multiplication (Charton, 2021; von Oswald et al., 2022). This is attributed to the increased number of layers and heads required in the inverse operation (Giannou et al., 2023). Nevertheless, gradient descent offers an alternative solution to linear regression, which requires only the matrix multiplication: $\boldsymbol{X}^T(\boldsymbol{Xw} - \boldsymbol{y})$, but is applied repeatedly.

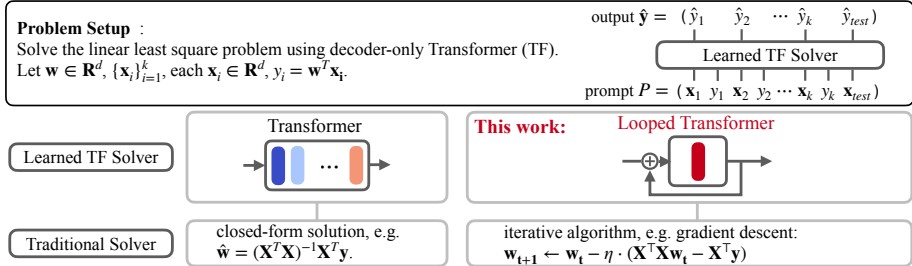

Figure 1: *How can a transformer be trained to learn an iterative learning algorithm?* Here we consider the task of training a transformer to solve linear regression in context. The provided prompt $(\boldsymbol{x}_1, y_1, \boldsymbol{x}_2, y_2, \cdots, \boldsymbol{x}_k, y_k, \boldsymbol{x}_{test})$ is fed into a decoder transformer. The objective is to reduce the squared loss between the predicted $\hat{y}_{\text{test}}$ based on this prompt, and the target value $f(\boldsymbol{x}_{\text{test}})$. Garg et al. (2022) demonstrated that a decoder transformer can learn to solve linear regression, which potentially involves learning the approximation of the least squares solution. In this study, we aim to train a transformer to learn iterative learning algorithms. Our goal is to achieve performance on par with standard transformers but with fewer parameters. To this end, we introduce the *looped* transformer architecture and its accompanying training methodology.

Motivated by this observation, we ask the following question:

> Can looped transformers emulate iterative learning algorithms more efficiently than standard, non-recursive transformers?

Within the specific function classes we tested, the answer is positive. Our preliminary findings on using *looped* transformer to solve the linear regression task are illustrated in Fig. 2. The remainder of the paper is organized as follows. In Sec. 4, we develop a training method for the *looped* transformer to emulate the desired performance of the iterative algorithm. Subsequently, in Sec. 5, we compare the empirical performance of the standard and the *looped* transformer and analyze the trade-off between them. Our contributions and findings are outlined below:

**Training methodology for Looped Transformer.** We propose a training methodology for looped transformers, aiming to effectively emulate iterative algorithms. The assumption for a *looped* transformer simulating a convergent algorithm is as loop iterations increase, the performance of the looped transformer should improve or converge. In alignment with this assumption, we delve into the structural design of the *looped* transformer, as well as investigate the number of loop iterations required during training. These investigations lead to the formulation of our training method.

**Performance of Looped Transformers for in-context Learning.** Based on our proposed training algorithm, empirical evidence demonstrates that *looped* transformer can be trained from scratch to in-context learn data generated from linear functions, sparse linear functions, decision trees, and 2-layer neural networks. Among the varied function classes examined, the *looped* transformer consistently outperforms the standard transformer, particularly when data is generated from sparse linear functions or decision trees. Our findings hint at the possibility that the looped transformer is more effective at in-context emulating learning algorithms, specifically for the learning tasks explored in this paper.

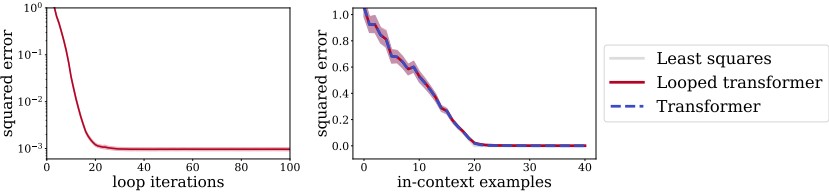

Figure 2: *The looped transformer can emulate iterative learning algorithms, offering performance comparable to standard transformers with reduced parameters.* We train a *looped* transformer to solve linear regression in-context. *(Left)*: While trained for 30 loop iterations, the *looped* transformer during inference achieves a stable fixed-point solution beyond the trained loop iterations. *(Right)*: The trained *looped* transformer matches the performance of a standard 12-layer transformer and closely aligns with the least squares solver, while using only 1/12 of the transformer's parameters.

## 2 RELATED WORKS

**Weight Sharing Models.** Weight sharing is inherent to recurrent models such as RNN (Bengio et al., 2003) and LSTM (Hochreiter & Schmidhuber, 1997). These sequence models reuse the weights in the direction of sequence processing, enabling possibly infinite lengths of sequence inputs. The transformer model (Vaswani et al., 2017; Devlin et al., 2019) typically has a fixed context length. To accommodate a longer context without increasing parameters, several works (Dai et al., 2019; Wang et al., 2019; Hutchins et al., 2022) employed transformers in a recurrent framework to enable an adaptive forward pass (Dai et al., 2019) or extend the context length (Hutchins et al., 2022). Moreover, several works (Lan et al., 2019; Jaegle et al., 2021; Dehghani et al., 2018) also proposed weight sharing along the forward pass of the model. This weight sharing is either across a limited number of layers, or with a halting criterion to exit the recurrence if needed. At one extreme is the implicit model (Bai et al., 2019), where the function is repeatedly applied an infinite number of times. The applications of these implicit models are addressed in the next paragraph.

**Deep Implicit Model.** Deep Implicit Models (Bai et al., 2018; 2019; 2020; Winston & Kolter, 2020; Bai et al., 2022) employ black-box solvers to find fixed-point solutions for implicit deep models. Later, Bai et al. (2021) proposed the Jacobian regularization to stabilize the training process. Nevertheless, this approach requires extensive hyperparameter tuning and still suffers from instability challenges. Implicit models have been demonstrated to solve math problems with extrapolation ability (Decugis et al., 2022). This is a special case of using the recurrent model to solve math tasks, as we will examine more closely in the next paragraph.

**Using Transformer to Solve Math or Reasoning Tasks.** Several works (Ongie et al., 2020; Doncevic et al., 2022; Zhang et al., 2022; Zhou et al., 2022b; Wei et al., 2022; Zhou et al., 2022a; Bansal et al., 2022; Charton, 2022; Goel et al., 2022; Zhang et al., 2023b; Dziri et al., 2023; Lee et al., 2023; Liu et al., 2022) have explored the ability of deep neural networks, including recurrent models, in solving mathematical or reasoning tasks that involve iterative processes. These tasks involve finite states or choices. Moreover, Zhang et al. (2023b) studied how transformers can emulate the structural recursion problem, while Ongie et al. (2020) investigated the use of recursive models in solving inverse problems in imaging. The regression problem, on the other hand, involves a continuous output space. Garg et al. (2022) investigated the transformer's ability to learn the continuous function classes. Akyürek et al. (2022) studied the phase transition of the solution obtained by transformers of varying depths when trained on linear regression and noisy linear regression problems. Raventos et al. (2023) studied the relationship between the number of tasks encountered during training and the performance of the transformer learning ridge regression problems in-context. Our work, however, utilizes the same architecture as in Garg et al. (2022)–a decoder model with non-linear attention–and aims to provide empirical evidence and insights into the learning dynamics of this model.

**Understanding How Transformer Learns In-Context to Solve Data-Fitting Problem.** In addition to approaching this problem from an empirical perspective, several studies have examined it from a theoretical standpoint, focusing on the construction of transformers and the linear attention network (excluding softmax counterparts). They connected the learnability of the transformer for the linear regression task with the implicit gradient descent updates on query prompt predictions (von Oswald et al., 2022; Ahn et al., 2023; Zhang et al., 2023a; Mahankali et al., 2023; Ding et al., 2023). von Oswald et al. (2022) empirically demonstrated that when minimizing the $\ell_2$ distance between predictions and true labels, one step of gradient descent is the optimal solution for a one-layer linear self-attention (LSA) transformer to learn when solving the linear regression problem, while Ahn et al. (2023); Zhang et al. (2023a; 2024) and Mahankali et al. (2023) provided theoretical explanations for it. Beyond 1-layer LSA, Fu et al. (2023); Giannou et al. (2024) have demonstrated that multi-layer transformers can perform higher-order optimization. Moreover, another line of work (Muller et al., 2021; Xie et al., 2021; Akyürek et al., 2022; Bai et al., 2023) provided a theoretical understanding of the transformer's in-context learnability through the lens of the Bayesian estimator. Li et al. (2023) studied the in-context learning from the viewpoint of generalization error. Lin & Lee (2024) investigated the dual operating modes of in-context learning.

## 3 PROBLEM SETTING

Let $\mathcal{F}$ denote a class of functions defined on $\mathbb{R}^d$. Let $\mathcal{D}_\mathcal{F}$ denote a probability distribution over $\mathcal{F}$ and let $\mathcal{D}_\mathcal{X}$ denote a probability distribution on $\mathbb{R}^d$. A random learning *prompt* $P$ is generated as follows. A function $f$ is sampled from $\mathcal{D}_\mathcal{F}$ and inputs $\{\boldsymbol{x}_1, \cdots, \boldsymbol{x}_k\}$ as well as the test sample $\boldsymbol{x}_{test}$ are sampled from $\mathcal{D}_\mathcal{X}$. The output of $\boldsymbol{x}_i$ is computed by $f(\boldsymbol{x}_i)$. The prompt is then $P = (\boldsymbol{x}_1, f(\boldsymbol{x}_1), \cdots, \boldsymbol{x}_k, f(\boldsymbol{x}_k), \boldsymbol{x}_{test})$ and the goal of a learning system is to predict $f(\boldsymbol{x}_{test})$. Let $M$

be a learning system and let $M(P)$ denote its prediction (note it is not given $f$ explicitly). The performance of $M$ is measured by the squared error $\ell(M(P), f(\boldsymbol{x}_{test})) = (M(P) - f(\boldsymbol{x}_{test}))^2$. In this work, we focus on transformer-based learning systems and compare them with other known learning systems depending on the tasks. Specifically, we examine the GPT-2 decoder model (Radford et al., 2019) with $L$ layers. By default, $L = 12$ for the unlooped transformer, following Garg et al. (2022)'s setting, and $L = 1$ for the looped transformer.

## 4    TRAINING ALGORITHM FOR LOOPED TRANSFORMER

In this section, we delve into the design choice for the algorithm-emulated looped transformer. For an algorithm-emulated looped transformer, we anticipate the following characteristics: 1) As loop iterations progress, the looped transformer should maintain or improve the performance; 2) the loop iterations have the potential to continue indefinitely without deterioration in performance.

**Training Strategy.**    Building on the problem setting in Sec. 3, let the prompt to the transformer be $P = (\boldsymbol{x}_1, f(\boldsymbol{x}_1), \cdots, \boldsymbol{x}_k, f(\boldsymbol{x}_k), \boldsymbol{x}_{test})$, with $P^i$ denotes the prompt prefix with $i$ in-context samples $P^i = (\boldsymbol{x}_1, f(\boldsymbol{x}_1), \cdots, \boldsymbol{x}_i, f(\boldsymbol{x}_i), \boldsymbol{x}_{i+1})$. The output of the *looped* transformer after $t$ looping iterations is
$$Y_t(P^i|\theta) = \underbrace{M_\theta(M_\theta(\cdots M_\theta(}_{t \text{ iterations}} Y_0^i + P^i) + P^i)\cdots + P^i)$$
where the transformer $M$ is parameterized by $\theta$, and $Y_0^i$ is a zero tensor with the same shape as $P^i$. Then we train the transformer by minimizing the following expected loss:

$$\min_\theta \mathbb{E}_P \left[ \frac{1}{b - b_0} \sum_{t=b_0}^{b} \frac{1}{k+1} \sum_{i=0}^{k} \big( Y_t(P^i|\theta), f(\boldsymbol{x}_{i+1}) \big) \right], \tag{1}$$

where we only measure the loss of the transformer over all prompt prefixes with loop iteration $b_0$ to $b$, with $b_0 = \max(b - T, 0)$. This truncated loss is inspired by the truncated backpropagation through time (Hochreiter & Schmidhuber, 1997; Hinton & Sutskever, 2013) for computation saving.

**Model Configuration and Parameter Count.**    For comparison with the unlooped transformer, we use a transformer identical to the one described in Garg et al. (2022), except for the number of layers. Specifically, we employ a GPT-2 model with an embedding dimension of $D = 256$ and $h = 8$ attention heads. The standard (unlooped) transformer has $L = 12$ layers, and the looped transformer has $L = 1$ layer. In terms of the number of parameters, the unlooped transformer comprises 9.48M parameters, and the looped transformer uses 0.79M. We follow the same training strategy: train with Adam optimizer, learning rate 0.0001, no weight decay or other explicit regularization (such as gradient clip, or data augmentation).

**Key Factors for Finding a Fixed-point Solution.**    When deploying this looped architecture training, two key factors come into consideration: 1) the input injection in the looped architecture (Sec. 4.1), and 2) the maximum loop iterations during training (Sec. 4.2). The subsequent sections will illustrate the impact of these two factors, using linear regression as the specific task for in-context learning.

### 4.1    INPUT INJECTION TO LOOPED TRANSFORMER

Reusing some notation, let $P$ represent the inputs to the transformer model $M(\cdot)$, and $Y_t$ be the output after applying $M(\cdot)$ for $t$ iterations. A looped transformer can be represented as $Y_{t+1} = M(Y_t; P), \forall t$. Several studies (Lan et al., 2019; Dehghani et al., 2018) have investigated a specific variant termed the *weight-tying* form: $Y_{t+1} = M(Y_t)$ with $Y_0 = P$, and $t < T < \infty$ for some constant $T$.

However, as $t$ approaches infinity, the influence of the initial input $Y_0$ diminishes, and the solution becomes essentially random or unpredictable (Bai et al., 2019; Bansal et al., 2022). To incorporate the input $P$ into the solution of the looped transformer, we propose setting $Y_{t+1} = M(Y_t + P)$, but the choice of injecting is not limited to addition. In Fig. 3, we display the outcomes when training a looped transformer either with (default) input injection or without (referred to as weight-tying). During inference, the weight-tying transformer will quickly diverge after the trained loop iterations.

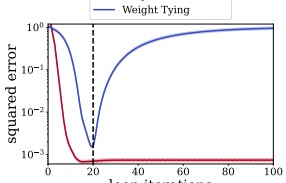

Figure 3: Test error for the linear regression problem using *looped* transformer, comparing models with default input injection to those without. Without input injection, the transformer's performance deteriorates beyond the trained loop iterations.

### 4.2    CHOICE OF LOOP ITERATIONS

To assess the *looped* transformer performance at the $b$-th loop iteration, we must execute this computation $b$ times consecutively. This can be analogized to a transformer architecture possessing $b$

layers in effect, albeit with shared weights throughout these layers. Choosing the value of $b$ requires a balance. A higher $b$ leads to longer training and inference time, whereas a lower $b$ limits the model's potential. Furthermore, as denoted in Eq. 1, the loss is truncated, with only the outputs of $T$ loop iterations contributing to the loss function.

Similarly, there's a trade-off when choosing $T$: a smaller $T$ results in reduced memory usage but negatively impacts performance, whereas a larger $T$ may cause the gradient to fluctuate, making the training process unstable. This section focuses on the optimal selection of $b$ and $T$ values (Fig. 4) for the linear regression task. We analyze the suitability of $b$ values within the set $\{12, 20, 30, 40, 50\}$, and $T$ values in $\{5, 10, 15\}$ for the linear function.

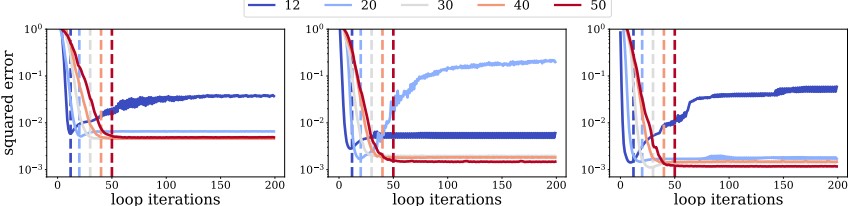

Figure 4: Evaluation of the *looped* transformer on in-context learning linear functions with different $b$ and $T$ during training ($b$ and $T$ are defined in Eq. 1). The figure from left to right is trained with $T = 5, 10, 15$, and different colors present different $b$ values (denoted in the legend). The solid lines of various colors depict how the looped transformer, trained with a specific value of $b$, performs as the loop iteration increases during inference. The corresponding dashed line represents the value of $b$.

Fig. 4 suggests that a lower $b$ value might cause the looped transformer to diverge after the trained loop iterations, leading to a less robust fixed-point solution. On the other hand, a higher $b$ value allows the model to locate a fixed-point solution that avoids divergence after the trained loop iterations. However, exceeding a certain $b$ value initiates a decline in the convergence rate of the loop iteration.

A similar trade-off applies to the selection of $T$; a lower $T$ may compromise performance, whereas a larger $T$ could induce instability in training, a phenomenon documented in the recurrent model training literature (Jaeger, 2005). A *looped* transformer with $b = 12$ matches the effective depth of the unlooped transformer studied in Garg et al. (2022). Nevertheless, it fails to replicate the performance of the standard (unlooped) transformer.

An additional observation is that the *looped* transformer consistently discovers a fixed-point solution that saturates prior to the trained iteration $b$. This saturation of the fixed-point solution occurs due to the loss objective, which requires the looped transformer to match the target within $b$ steps. Consequently, selecting a smaller value of $b$ value expedites convergence to the fixed-point solution. However, beyond a certain value of $b$, the convergence rate reaches saturation regardless of the increase in $b$. For instance, training with $T = 15$, $b = 40, 50$ yields similar convergence rates.

**Optimal Choice of $b$ and $T$.** Our goal is to efficiently train the *looped* transformer for the in-distribution task. This requires determining the minimal $b$ value that prevents divergence after training loop iterations. To minimize memory usage, we prefer a smaller $T$ value. Guided by these criteria, we adopt $b = 20$ and $T = 15$ for the linear regression task.

## 5 EXPERIMENTAL RESULTS

### 5.1 EXPERIMENTAL SETUP

We focus on the data-fitting problems generated by the linear model with problem dimension $d = 20$, and in-context sample $k = 40$. The parameters are sampled from $\mathcal{N}(0, I_d)$, and the in-context samples $x_i \sim \mathcal{N}(0, I_d)$ as well. When measuring the performance, we evaluate 1280 prompts and report the 90% confidence interval over 1000 bootstrap trials. To ensure the error is invariant to the problem dimension, we also do normalization to the error as specified in Garg et al. (2022).

**Scheduled Training.** We follow the training curriculum in terms of $d$ and $k$, as specified in Garg et al. (2022). Besides, we also implement a schedule on the parameter $b$, where $b$ is progressively increased during the course of training. As a consequence, the truncated loss window starts at $[1, T]$ and then shifts over time to $[b - T, b]$. It is worth noting that minimizing the mean squared error between the $[1, T]$ loop iteration and the target can be more challenging than for $[b - T, b]$.

Consequently, we avoid referring to this approach as "curriculum learning," as that term typically refers to starting with a less difficult task and gradually moving on to more challenging ones. Rather,

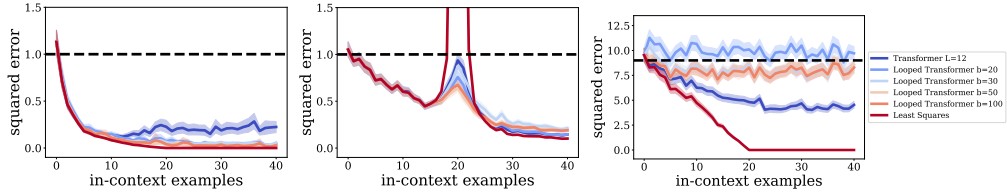

(a) Skewed Covariance     (b) Noisy Linear Regression     (c) Scaled $x$ with scaling=3

Figure 6: Evaluation of the transformer trained on linear functions, tested under various conditions: a) inputs with skewed covariance, b) noisy linear functions, and c) scaled inputs.

this strategy can be seen as a method to kick-start the training of the transformer model in a loop structure. In general, the decision to use or not to use the scheduling does not significantly impact the outcome. However, in some instances, we observe superior results achieved by training with a schedule for $b$. More details can be found in Appendix E.

## 5.2 LOOPED TRANSFORMER CAN IN-CONTEXT LEARN LINEAR REGRESSION

***Looped* Transformer Matches the Performance of Standard Transformer when Evaluating In-Distribution.** As indicated in Fig. 2 (right), the *looped* transformer trained with $b = 20$ and $T = 15$ is able to match the performance of the standard transformer, almost matching the performance of the traditional optimal solver: the ordinary least squares.

***Looped* Transformer Learns Efficiently with Fewer Distinct In-Distribution Prompts.** We further evaluate the sample complexity of transformer training. Specifically, we ask how many distinct prompts need to be seen during training for the transformer or the *looped* transformer to learn the function. To do so, instead of generating the linear function on the fly in each iteration, we generate the training dataset before training. Thus, during training, the transformer may encounter the same prompt multiple times.

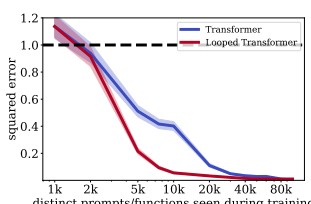

Figure 5: Performance of transformer on linear functions with $d = 10$ and $k = 21$, when trained with different numbers of distinct prompts/functions.

To isolate the impact of curriculum learning, we disable this strategy during and focus on the linear regression task with problem dimension $d = 10$, in-context samples $k = 20$. The results are presented in Fig. 5. Due to the fewer parameters in the looped transformer, it is able to learn the function with fewer distinct prompts/functions compared to the standard transformer. More results regarding different problem dimensions are presented in Appendix C.

***Looped* Transformer Exhibits a Certain Inductive Bias when Evaluating Out-of-Distribution.** Recall that the transformer is trained with $x \sim \mathcal{N}(0, I_d)$. We now evaluate it on: a) inputs with skewed covariance, b) inputs with noise, and c) in-context example inputs and query inputs that lie in different orthants. The result is shown in Fig. 6. Rather than mastering the true algorithm applied to the linear regression problem irrespective of shifts in the input distribution, the *looped* transformer exhibits an inductive bias favoring simpler solutions when compared to the unlooped transformer. This results in improved handling of (a) skewed covariance inputs.

In the task of (b) noisy linear regression, the *looped* transformer displays a reduced peak in the double descent curve, similar to the effects of applying minor regularization to the base model for resolving noisy linear regression. Bhattamishra et al. (2022) also reported the simplicity bias of the random transformer towards low sensitivity in the boolean function base. However, this inductive bias towards simplicity can negatively impact performance when there is a scaling shift in the input distribution. As depicted in Fig. 6 (c), during the inference process, if we sample $x$ such that $x = 3z$, and $z \sim \mathcal{N}(0, I)$, the *looped* transformer underperforms compared to the unlooped transformer. More extrapolation results can be found in Appendix D.

## 5.3 IMPACT OF MODEL ARCHITECTURE VARIATIONS ON LOOPED TRANSFORMER PERFORMANCE

In this section, we explore the impact of varying the number of layers ($L$), heads ($h$), and embedding dimension ($D$) on the *looped* transformer. These experiments are trained with $b = 30$, $T = 15$.

**Varying the Number of Layers.** In Fig. 7 (left), we plot the squared error for the transformer with $L$ layers, and the looped transformer with $L$ layers, applying $t$ loop iterations. From the figure, we observe that as $L$ increases, convergence is achieved more rapidly. To match the performance of a standard transformer with $L$ layers, the looped transformer needs more than $L$ loop iterations, i.e. the

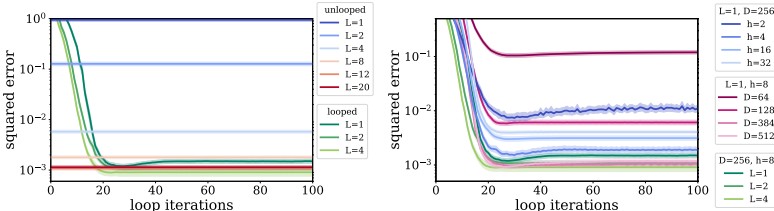

Figure 7: Let $D$ be the transformer embedding dimension, $L$ be the number of layers, and $h$ be the number of heads. For linear functions with problem dimension $d = 20$, and in-context samples $k = 40$, we test the following: (left) the standard (unlooped) transformer and *looped* transformer with $D = 256$ and $h = 8$, but varying $L$, and (right) *looped* transformer with varying $L$ and $D, h$.

effective depth of the looped transformer exceeds that of the standard transformer. For instance, a looped transformer with a single layer requires roughly 20 iterations in order to achieve an 8-layer transformer's performance. This suggests that the transformer and the *looped* transformer learn different representations at each layer (iteration).

To delve deeper into the learning dynamics of each layer in both the transformer and the looped transformer, we employ the model probing technique suggested by Akyürek et al. (2022); Alain & Bengio (2016). Reusing some notation, we represent the output of the standard transformer's $t$-th layer or the looped transformer's $t$ loop iterations as $Y_t$. An MLP model is trained on $Y_t$ to learn target probes, specifically $\boldsymbol{X}^T\boldsymbol{y}$ for the gradient component and the $\boldsymbol{w}_{ols}$ for the optimal least squares solution. The mean squared error for model probing is illustrated in Fig. 8. While standard transformers initially capture representations relevant to $\boldsymbol{X}^T\boldsymbol{y}$ and $\boldsymbol{w}_{ols}$, and subsequently shift focus to other statistics, the *looped* transformer consistently refines its representations related to the target probes across its loop iterations. Details of the model probing are presented in Appendix F.

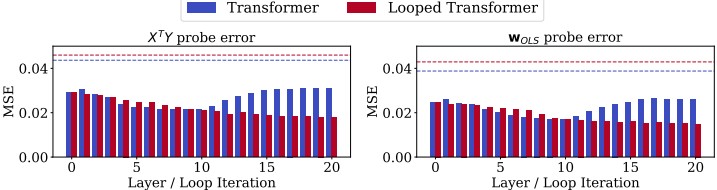

Figure 8: *How do the transformer and the looped transformer encode information across layers/iterations?* We train a 2-layer MLP probing model to recover the target probe from the transformer's output at $t$-th layer/loop iteration. The dashed line indicates the minimal probe error obtained in a control task, where the linear regression parameter $\boldsymbol{w}$ is fixed to be $\mathbf{1}$. Contrasting with the standard transformer, which decodes the target probe optimally around the 10-th layer, the *looped* transformer steadily refines its representation, improving the decoding of the target probe with each subsequent loop iteration.

**Varying the Number of Heads and Embedding Dimension.** Fig. 7 (right) illustrates the squared error of the *looped* transformer for various combinations of $L$, $D$, and $h$. With $L = 1$ and $h = 8$ held constant, increasing $D$ leads to improved convergence, saturating at $D = 256$. When holding $D = 256$ and $h = 8$ constant and varying $L$, the *looped* transformer shows similar convergence performance. The primary differences are in the speed of convergence. Adjusting only $h$ with fixed $L$ and $D$, the looped transformer's error decreases as $h$ rises from 2 to 8. However, the error increases when $h$ progresses from 8 to 32. Here, we follow the standard implementation of GPT-2[2], where the per-head embedding dimension is defined as $\frac{D}{h}$. This definition implies a trade-off in the transformer between the number of heads and the per-head embedding dimension as $h$ varies.

## 5.4 HIGHER COMPLEXITY FUNCTION CLASSES

In this section, we shift our focus to function classes of higher complexity: a) the sparse linear model with $d = 20$, $k = 100$, and non-sparse entry $s = 3$; b) the decision tree with depth$= 4$ and input dimension $d = 20$; c) the 2-layer ReLU neural network with 100 hidden neurons and input dimension $d = 20$. For these functions, parameters are sampled from $\mathcal{N}(0, I_d)$, and in-context samples $\boldsymbol{x}_i \sim \mathcal{N}(0, I_d)$. This setup follows the methodology described in Garg et al. (2022), and is further detailed in Appendix A. We also conduct experiments on OpenML datasets, which better represents practical scenarios. Details of experiment setup and results for OpenML (Vanschoren et al., 2013) datasets are available in Appendix G.

**Optimal Loop Iterations in the Looped Transformer for Different Functions.** In Sec. 4.2, we highlight that for the *looped* transformer trained on a linear regression task, a larger value of $b$ and an

---

[2]https://github.com/karpathy/nanoGPT

appropriate $T$ enhance its ability to discover a fixed-point solution that remains stable beyond the trained loop iterations. This observation is consistent across various functions, but the optimal $b$ and $T$ values may vary. Fig. 9 depicts the performance of *looped* transformer trained with $T = 10$ but varying $b$ values on different functions.

Comparison with Fig. 4 indicates that tasks with higher complexity, like linear regression (which is more parameter-intensive than sparse linear regression), necessitate increased $b$ and $T$ values to achieve stable convergence during training. For instance, with $T = 10$, linear regression diverges at $b = 20$, whereas sparse linear regression converges to a fixed point. The values of $b$ and $T$ required can be indicative of the complexity a transformer faces when tackling a specific task. Interestingly, while learning a 2-layer neural network is more challenging (Chen et al., 2022), it requires fewer $b$ and $T$ for the looped transformer in comparison to seemingly simpler tasks, such as linear regression. Complete results of the looped transformer trained with various $b$ and $T$ are presented in Appendix B.

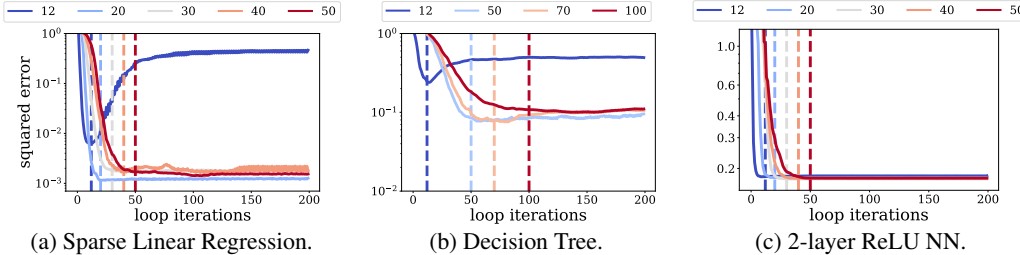

(a) Sparse Linear Regression.   (b) Decision Tree.   (c) 2-layer ReLU NN.

Figure 9: Evaluation of the *looped* transformer on in-context learning data generated from a) the sparse linear function, b) the random decision tree, and c) the 2-layer ReLU neural network with $T = 10$ and different $b$ during training ($b$ is defined in Eq. 1). The solid lines of various colors depict how the looped transformer, trained with a specific value of $b$, performs as the loop iteration increases during inference. The corresponding dashed line represents the value of $b$.

**Looped Transformer Matches or Outperforms Standard Transformer.**   Through a comprehensive hyperparameter search, we identify optimal values for $b$ and $T$, as detailed in Appendix B. Specifically, for data generated from a) sparse linear functions, we use $b = 20$ and $T = 10$; b) decision trees, $b = 70$ and $T = 15$; and c) 2-layer ReLU NNs, $b = 12$ and $T = 5$. We then compare the performance of *looped* transformer against the standard transformer and other baseline solvers, as depicted in Fig. 10. Of all the tasks we examine, the *looped* transformer consistently matches the standard transformer, except for the sparse linear regression tasks, where the *looped* transformer **outperforms** the standard transformer, and even the Lasso (Tibshirani, 1996). By **outperforming**, we do not refer to the precision of the final solution but to the trend relative to the number of in-context samples. For instance, with 40 in-context samples, the Lasso achieves a solution with an error of 1.32e-04, the Least Squares reaches 8.21e-14, the looped transformer achieves 6.12e-04, and the standard transformer achieves 0.0017. However, with fewer than 10 in-context samples, the looped transformer has a smaller error than the Lasso solution. Here we do a hyperparameter search for Lasso over $\alpha \in \{1e - 4, 1e - 3, 0.01, 0.1, 1\}$, where $\alpha$ is the $\ell_1$ parameter, and select the best $\alpha = 0.01$. We conjecture that this performance gain is a result of the inductive bias of the *looped* transformer, which favors simpler (i.e., sparser) solutions over their more complex counterparts. When tackling sparse problems, this inherent bias enhances the performance of the *looped* transformer.

**Performance Analysis of Looped Transformer Across Varying Sparsity Levels in Sparse Linear Regression.**   The looped transformer demonstrates enhanced performance in the sparse linear regression task compared to the unlooped transformer. To further investigate this, we examine the task under varying levels of sparsity, which represent different task difficulties. To enable the *looped* transformer to handle all complexity levels, we choose parameters $b = 30$ and $T = 15$. As presented

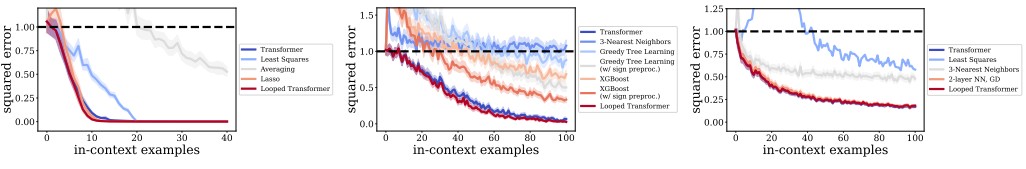

(a) Sparse Linear Regression.   (b) Decision Tree.   (c) 2-layer ReLU NN.

Figure 10: Performance evaluation of the trained transformer on in-context learning data from: a) sparse linear functions, b) random decision trees, and c) 2-layer ReLU neural networks. The *looped* transformer matches or even surpasses the transformer's performance using only 1/12th of the parameters employed by the latter.

in Fig. 11, the results show that across all sparsity levels of the weight vector for the sparse linear function, the *looped* transformer consistently outperforms the unlooped transformer, especially when the number of in-context samples is limited.

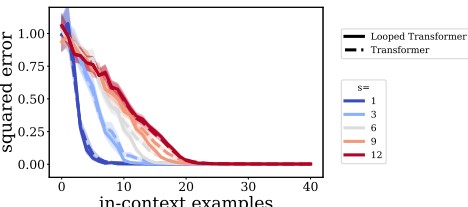

Figure 11: Performance of *looped* transformer in solving sparse linear functions with problem dimension $d = 20$. We examine sparsity levels $s = 1, 3, 6, 9, 12$, where only $s$ entries are non-zero. Across all sparsity levels, the *looped* transformer consistently matches or outperforms the unlooped transformer, achieving more accurate solutions with fewer in-context samples.

## 6 DISCUSSION AND FUTURE WORK

**Mathematical Insights of Looped Transformers.** The expressive power of recurrent models has been widely investigated in the literature: Theorem 3 in Bai et al. (2019) demonstrated that any traditional $L$-layer neural network can be represented by a weight-tied, input-injected network. Further, Giannou et al. (2023) explored the potential of an encoder looped transformer to function as a universal computer. Our work concentrates on empirically validating the looped architecture, offering a practical training method for a looped decoder transformer in in-context learning. Investigating the expressiveness of this transformer presents a promising future research direction.

**Looped Transformer Emulates Fixed-Point Iteration Method Within Training Distribution.** Through our designed training method, the looped transformer successfully approximates fixed-point solutions beyond the iterations it was trained on, effectively emulating the fixed-point iteration method within the training distribution. Nevertheless, these solutions come with an inherent error floor and have limited performance on out-of-distribution prompts. As a result, our trained looped transformer fails to identify an actual algorithm that generalizes to any input distribution, but only emulates the algorithm with the particular input distribution it is trained on. Overcoming this limitation, and enhancing the transformer-derived solver to match the generalization capabilities of conventional algorithms presents a promising avenue for future research.

**Choice of $b$ and $T$ and the Notion of Task Difficulty.** As discussed in Sec. 5.4, increasing the loop iterations ($b$) and truncated loss window size ($T$) enables the looped transformer to find a fixed-point solution that does not diverge beyond the trained loop iterations. The optimal values for $b$ and $T$ are determined through hyperparameter tuning, selecting the minimum parameters necessary to prevent divergence in the looped transformer. These optimal values of $b$ and $T$ reflect the complexity of a task from the transformer's perspective. A task that is inherently more complex will require larger values of $b$ and $T$ to achieve convergence to a fixed-point solution, whereas simpler tasks will require smaller values. Thus, the parameters $b$ and $T$ could be used as a metric to assess the difficulty level of tasks for transformers. Expanding on this, in practice where tasks exhibit varying degrees of complexity, an adaptive looping strategy could be beneficial. These strategies could include token-level adaptive loop iterations, as proposed by Dehghani et al. (2018), or in-context multi-task inference.

**Choice of $b$ and $T$ and their Memory-Computation Trade-off.** The computational cost of training a looped transformer is tied to the selected values of $b$ and $T$. Specifically, $b$ determines the transformer's forward pass time, since the looped transformer has to assess $b$-th loop iteration's output by unrolling $b$ times. $T$ influences the memory requirements during training, given that the loss is computed over $T$ iterations. If the goal is to approximate a fixed-point solution using the looped transformer, one could circumvent intensive tuning of $b$ and $T$. Instead, opting for a larger $b$, a suitably chosen $T$, and incorporating regularization strategies such as mixed-precision training, gradient clipping, and weight decay may be beneficial. These regularization strategies could potentially alleviate the instability issues associated with $T$. We omit these regularization methods to maintain a fair comparison with the transformer training methodologies outlined in Garg et al. (2022).

## 7 CONCLUSION

Motivated by the widespread application of iterative algorithms in solving data-fitting problems across various function classes, we analyzed the loop architecture choice and training strategy, then developed a training method for the looped transformer. This methodology enables the looped transformer to approximate fixed-point solutions beyond the initially trained loop iterations. We hope this work can contribute to the community in understanding and developing better training strategies for transformers in solving data-fitting problems.

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

## A  DETAILED SETUP FOR FUNCTION CLASSES

For completeness, we detail the setup of the function classes here, adhering to the settings in Garg et al. (2022). Inputs $x \sim \mathcal{N}(0, I)$ are consistent across all function classes.

**Linear Function.**  For linear function with dimension $d$, we sample the parameter of linear function $w \sim \mathbb{R}^d$ following $\mathcal{N}(0, I)$.

**Sparse Linear Function.**  We follow the same distribution setting as in linear function, and we uniformly select $s$ out of $d$ dimension to be nonzero.

**Decision Tree.**  We employ a binary decision tree with a depth of 4. At each non-leaf node, the decision to branch left or right is based on the sign of a randomly selected coordinate of the input $x$. This coordinate is chosen uniformly at random. The leaf nodes are initialized following a normal distribution $\mathcal{N}(0, I)$. The evaluation of an input $x$ involves traversing from the root to a leaf node, moving left for a negative coordinate and right for a positive one, with the output being the value of the reached leaf node.

**2-layer ReLU NN.**  We initialize the weights of the network following $\mathcal{N}(0, I)$, and set the hidden layer dimension to be 100.

## B  CHOICE OF LOOP ITERATIONS AND ITS EFFECT ON IN-CONTEXT PERFORMANCE

In this section, we present the result (Fig. 12) of the *looped* transformer performance when trained with different $b$ and $T$ for the following function classes: a) linear functions with problem dimension $d = 20$ and in-context samples $k = 40$, b) sparse linear functions with problem dimension $d = 20$, in-context samples $k = 40$, and non-sparse entry $s = 3$, c) decision tree with problem dimension

$d = 20$, depth to be 4, and in-context samples $k = 100$, as well as d) 2-layer ReLU NN with problem dimension $d = 20$, 100 hidden neurons, and in-context samples $k = 100$.

Recall that the target function we aim to minimize during the training of *looped* transformer is:

$$\min_{\theta} \mathbb{E}_P \left[ \frac{1}{b - b_0} \sum_{t=b_0}^{b} \frac{1}{k+1} \sum_{i=0}^{k} \left( Y_t(P^i|\theta), f(\boldsymbol{x}_{i+1}) \right) \right],$$

where $P^i$ denotes the prompt prefix with $i$ in-context samples $P^i = (\boldsymbol{x}_1, f(\boldsymbol{x}_1), \cdots, \boldsymbol{x}_i, f(\boldsymbol{x}_i), \boldsymbol{x}_{i+1})$, $Y_t(P^i|\theta)$ is the output of the *looped* transformer with parameter $\theta$ after $t$ looping iterations. $b_0 = \max(b - T, 0)$. The choice of $b$ affects how many loop iterations the *looped* transformer needs to unroll during the training, and $T$ represents the truncated window size for the loss calculation.

For linear regression, sparse linear regression, and 2-layer ReLU NN tasks, we investigate the different values of $b$ in $\{12, 20, 30, 40, 50\}$, and for the more challenging task – decision tree, we search over $b$ values in $\{12, 50, 70, 100\}$. For all tasks, we investigate $T$ values in $\{5, 10, 15\}$. Larger $T$ values such as $T = 20$ will result in training instability if trained without any stabilization technique such as weight decay, mixed-precision, or gradient norm clip. But once employed with those techniques, it is safe to train with large $T$ as well. In Fig. 12 and Fig. 13, we present the result of the looped transformer, when trained with different $b$ and $T$, the performance on different tasks, with respect to the loop iterations, and with respect to the in-context samples.

For each task, the results are consistent: training with smaller $T$ will yield a larger mean squared error in the in-context learning solution. When $T$ is held constant, larger $b$ will enable the *looped* transformer to find a fixed-point solution. As $b$ increases, the performance improves until it reaches a point of saturation.

Across tasks, we can see a clear pattern of the choice of $b$ and $T$ related to the level of difficulty of the task. Sparse linear regression, which has fewer parameters in its function class, requires fewer $b$ and $T$ to find a converged solution compared to linear regression. The decision tree task requires the learning of $2^{\text{depth}+1} - 1$ number of parameters (in our case, 31 parameters). Thus it requires a larger $b$ to find a fixed-point solution. Counterintuitively, the 2-layer ReLU neural network, which contains in total $21 \times 100$ number of parameters, only requires $b = 12$ to find a fixed-point solution. The reason why data generated from a 2-layer ReLU neural network only requires a small value of $b$ to find the fixed-point solution remains unclear. We conjecture that the value of $b$ is an indication of the task's difficulty for the looped transformer to solve.

## C  EFFECT OF NUMBER OF DISTINCT PROMPTS / FUNCTIONS SEEN DURING TRAINING

In Section 5, we investigated the impact of the number of distinct prompts during transformer training for a problem dimension $d = 10$ and in-context samples $k = 20$. In this section, we extend the results to include (1) $d = 5$, $k = 10$, and (2) $d = 20$, $k = 40$. It is important to note that all experiments were conducted without the use of curriculum learning for $d$ and $k$. For the looped transformer, we set $b = 30$, and $T = 15$, and we don't apply scheduling in the loop iterations as well. For $d = 5$, we investigated learning rate in $\{0.0001\}$, for $d = 10$ with learning rate $\{0.00001, 0.0001\}$, and for $d = 20$, we explored learning rate $\{0.00001, 0.0001, 0.001\}$, selecting the result with the best performance. The result is presented in Fig. 14. Across different problem dimensions, the looped transformer demonstrated better sample efficiency compared to the vanilla transformer.

## D  EXTRAPOLATION BEHAVIOR OF LOOPED TRANSFORMER AND UNLOOPED TRANSFORMER

We follow the setup from Garg et al. (2022) on testing the out-of-distribution behavior. Recall that both the transformer and the looped transformer are trained to solve linear regression tasks in-context. During training, the inputs are sampled from $\mathcal{N}(0, I_d)$, and the linear function weights $\boldsymbol{w} \sim \mathcal{N}(0, I_d)$. Here $d$ is 20, and the number of in-context samples is $k = 40$. We now evaluate these models on: a) inputs with skewed covariance; b) inputs that lie in a $d/2$-dimensional subspace; c) inputs with noise;

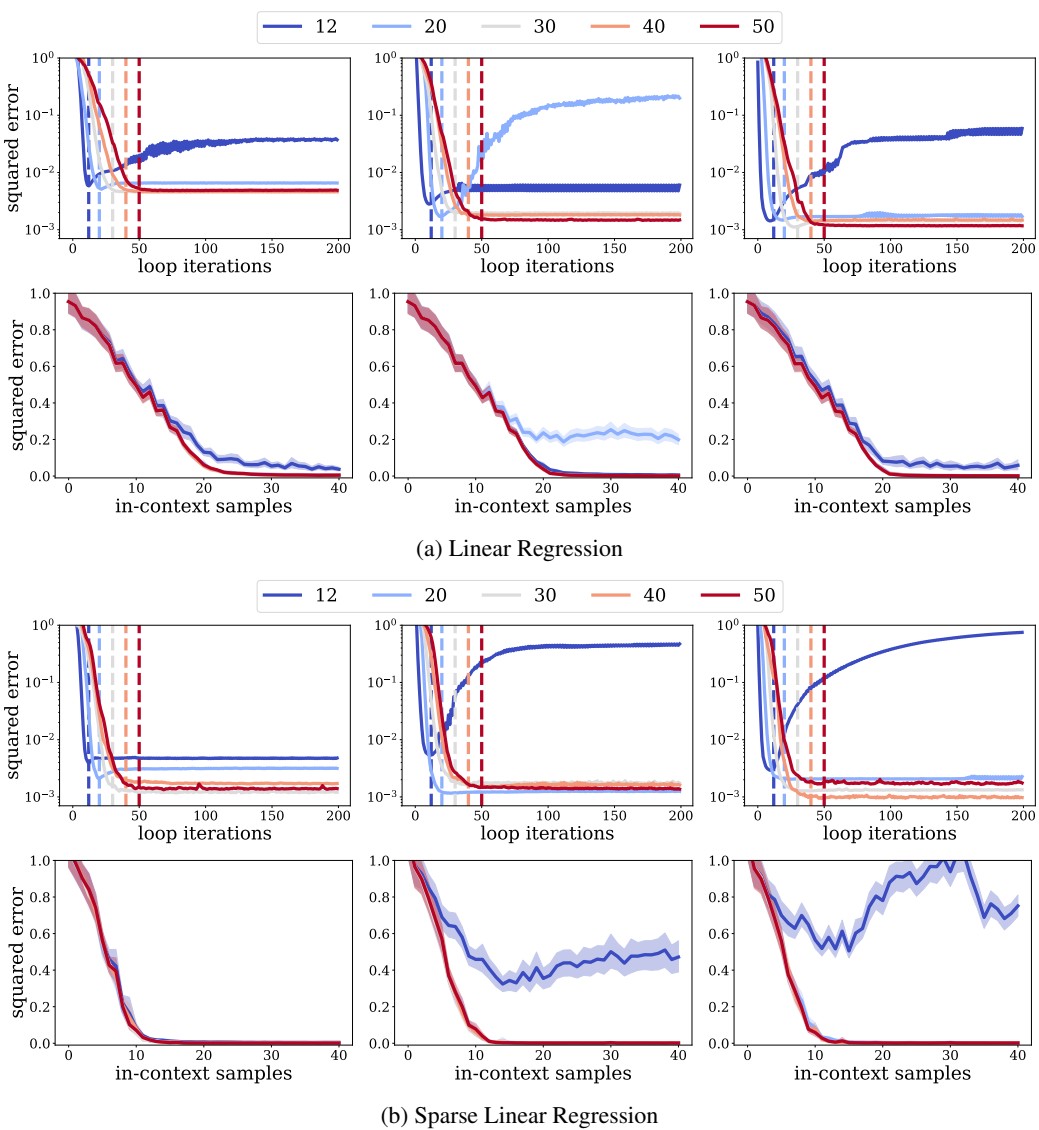

Figure 12: We evaluate the *looped* transformer on in-context learning of a) linear functions, and b) sparse linear functions with different $b$ and $T$ during training ($b$ and $T$ is defined in Eq. 1). For each block of figures, from left to right is with $T = 5, 10, 15$. (top) Performance with respect to different loop iterations, (bottom) performance with respect to different numbers of in-context samples.

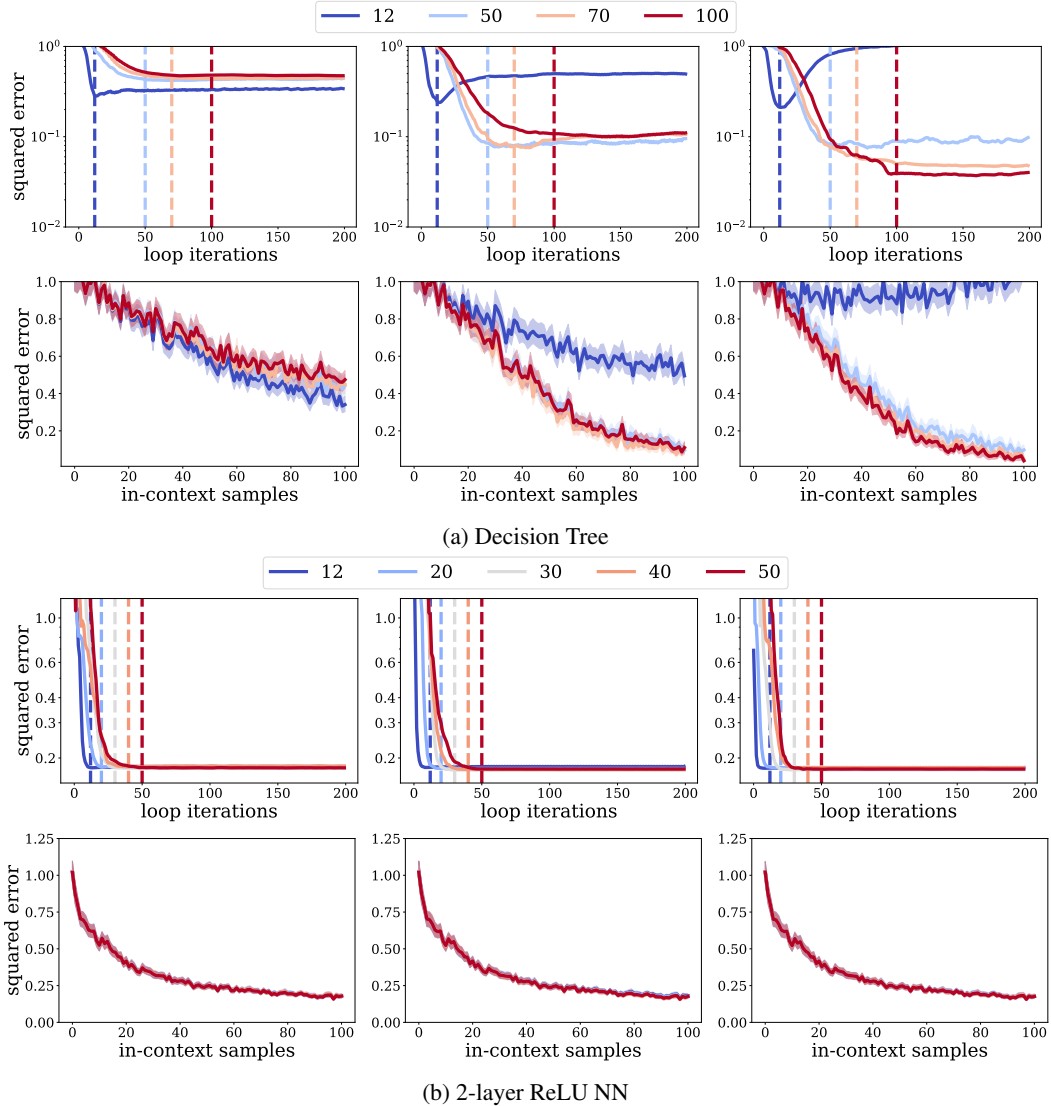

(a) Decision Tree

(b) 2-layer ReLU NN

Figure 13: We evaluate the *looped* transformer on in-context learning of a) random decision tree, and b) 2-layer ReLU neural network with different $b$ and $T$ during training ($b$ and $T$ is defined in Eq. 1). For each block of figures, from left to right is with $T = 5, 10, 15$. (top) Performance with respect to different loop iterations, (bottom) performance with respect to different numbers of in-context samples.

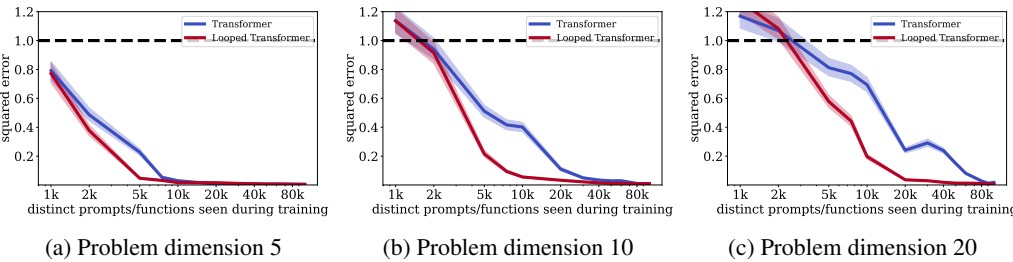

(a) Problem dimension 5      (b) Problem dimension 10      (c) Problem dimension 20

Figure 14: Performance of the standard transformer and the looped transformer on linear regression problem with problem dimension $d = 5, 10, 20$, when trained with a different number of distinct prompts/functions.

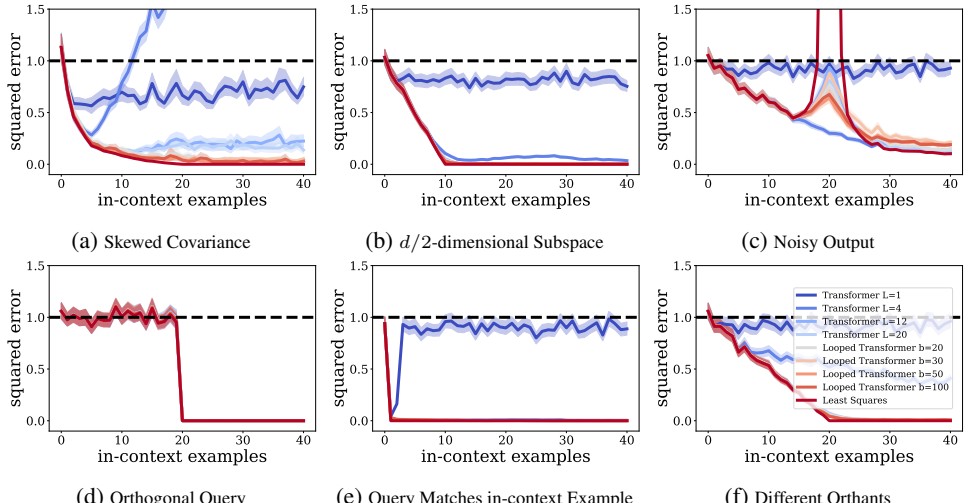

Figure 15: Evaluating the transformer of the different number of layers $L$, and looped transformer, with the number of layer to be $L = 1$, trained to maximum $b$ loop iterations during training. The transformer is trained on the linear regression task, but evaluated on different out-of-distribution tasks.

d) the query is orthogonal to the in-context example; e) the query is equal to one of the in-context samples; and f) the in-context example inputs and query inputs lie in different orthants. The results are presented in Fig. 15.

From the figure, it is evident that the looped transformer, in some out-of-distribution cases, exhibits performance similar to the standard transformer with fewer layers. For instance, in the noisy linear regression case, the transformer with a number of layers $L = 4$ exhibits a reduced peak when $d$ in-context samples are presented. This trend is similar to the looped transformer, especially when the $b$ value is larger. In certain other cases, the looped transformer demonstrates performance akin to the standard transformer with a higher number of layers. For instance, when the prompts are (a) with skewed covariance, (b) in $d/2$-dimensional subspace, or (f) the in-context samples and the prompts lie in different orthants, for the standard transformer, the larger $L$ is, the better the performance is. Moreover, the looped transformer can match or even outperform the standard transformer with as many as $L = 20$ number of layers. Recall that for the linear regression task, when $b < 20$, the looped transformer cannot find a fixed-point solution. Therefore, we didn't include the looped transformer with $b < 20$ in the comparison. In other cases, the looped transformer performs similarly to the standard transformer. This is evident in instances such as (d) when the in-context samples and the query lie in orthogonal space, and (e) when the query matches one of the in-context samples.

We further evaluated the out-of-distribution performance of the looped transformer by either scaling the prompt $x$ or scaling the parameter of the linear function $w$ during inference. The results are depicted in Fig. 16. As illustrated in the figure, the looped transformer demonstrates a performance comparable to that of the standard transformer with fewer layers. For instance, when $x$ is scaled by 2, the looped transformer performs similarly to the standard transformer with $L = 4$.

## E  EFFECTS OF SCHEDULING

In Sec. 5.1, we discuss the training schedule for the loop iterations – we will gradually increase $b$ during the course of training. As a result, the truncated loss window starts at $[1, T]$, and then gradually shifts to $[b - T, b]$. In this section, we investigate the impact of this scheduling during training. Specifically, we compare the looped transformer training on the four function classes: a) linear function; b) sparse linear function; c) decision tree; d) 2-layer ReLU neural network, when trained with and without the scheduling. When training, we adhere to the selected $b$ and $T$ as indicated in Sec. 5.4. Results are presented in Fig. 17. As indicated in the figure, in most cases, the looped transformer's performance does not differ when trained with or without the scheduling. However, for

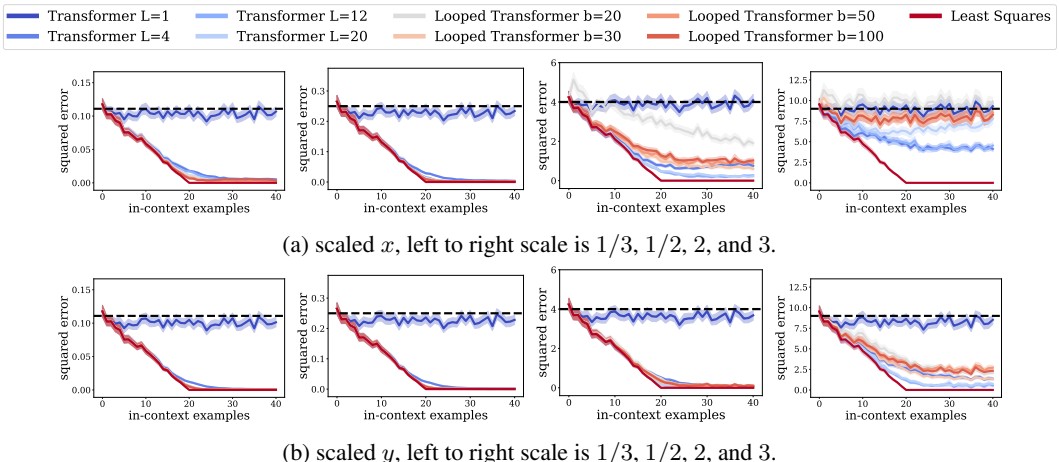

(a) scaled $x$, left to right scale is $1/3$, $1/2$, 2, and 3.

(b) scaled $y$, left to right scale is $1/3$, $1/2$, 2, and 3.

Figure 16: Evaluating the transformer of the different number of layers $L$, and looped transformer, with the number of layer to be $L = 1$, trained to maximum $b$ loop iterations during training. The transformer is trained on the linear regression task, but evaluated on scaled prompt $x$ or scaled linear function parameter $w$.

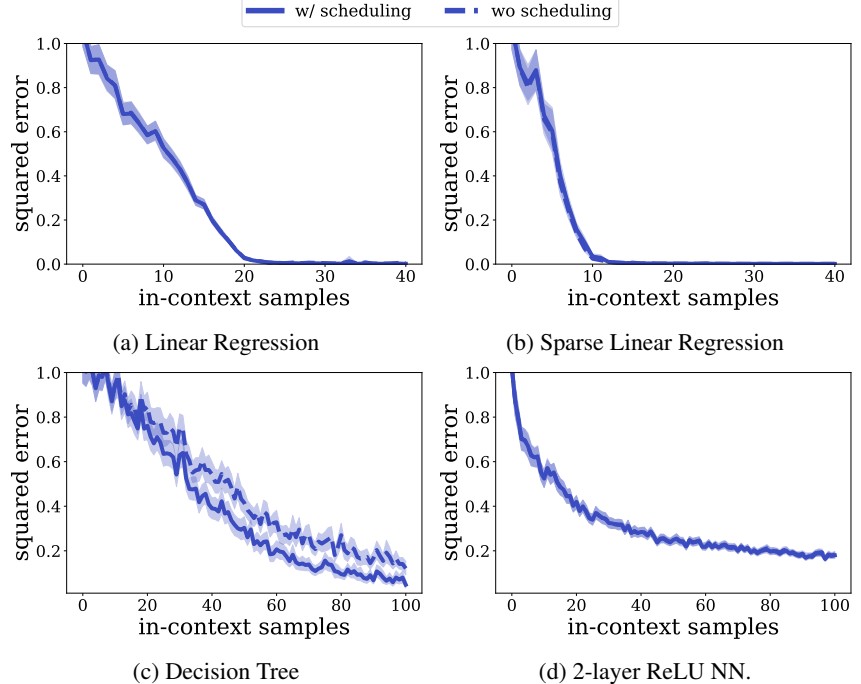

(a) Linear Regression

(b) Sparse Linear Regression

(c) Decision Tree

(d) 2-layer ReLU NN.

Figure 17: Performance of the looped transformer with respect to a varying number of in-context samples. We compared the trained looped transformer in solving (a) linear regression, (b) sparse linear regression, (c) decision tree, and (d) 2-layer ReLU NN, when trained with (solid line) or without (dashed line) scheduling. Whether scheduling is applied over $b$ has impacts on the performance when training the *looped* transformer on data generated from the decision tree, with the application of scheduling resulting in slightly better performance.

the decision tree task, the looped transformer trained with scheduling exhibits better performance compared to when trained without scheduling. Therefore, throughout our experiments, we opt to use scheduling for the loop iterations.

## F    DETAILS OF MODEL PROBING

In Sec. 5.3, we analyze the output of either the standard transformer's $t$-th layer or the looped transformer's $t$-th loop iteration by training a probing model on them. In this section, we present the details of the model probing experiments.

Reusing the notation, we denote the output at the $t$-th layer or loop iteration as $Y_t$. $Y_t$ is of shape $k \times D$, where $k$ represents the sequence length, and $D$ represents the embedding dimension of the transformer. In our experiments, we investigate two target probes: (a) $\boldsymbol{X}^T\boldsymbol{y}$, representing the gradient component, and $\boldsymbol{w}_{ols}$, representing the optimal least square solution. Given the problem dimension is $d$, the target probes $\boldsymbol{v}$ are of shape $d \times 1$. For simplicity, we omit the batch size.

Before feeding the representation $Y_t$ into the probing model, we first compute a weighted sum over the sequence dimension. Given $\boldsymbol{s} \in \mathbb{R}^{k \times 1}$, we perform the following operations:

$$\alpha = \text{softmax}(\boldsymbol{s}) \tag{2}$$
$$Z_t = \alpha^T Y_t \tag{3}$$

With $Z_t \in \mathbb{R}^{1 \times D}$, we employ a multi-layer perceptron model (MLP) $f_{\text{MLP}}$ to probe the output. The MLP model consists of two linear layers, with ReLU in between, with $d_{\text{hidden}}$ denote the number of hidden neurons in MLP:

$$\hat{\boldsymbol{v}} = f_{\text{MLP}}(Z_t)$$

When training the $\boldsymbol{s}$ and $f_{\text{MLP}}$, we aim to minimize the mean squared error between $\hat{\boldsymbol{v}}$ and $\boldsymbol{v}$. For each layer's output, and each in-context length, we train a distinct $\boldsymbol{s}$ and $f_{\text{MLP}}$ to probe the target. When presenting the results, we compute the average loss over various context lengths. Additionally, to normalize the error, we divide the mean squared error by the problem dimension $d$.

In our experiments, we set $D = 256$, $d = 20$, and $d_{\text{hidden}} = 64$. We optimize the probing model with Adam optimizer and learning rate $0.001$, which is found to be the best among the set $\{0.01, 0.001, 0.0001\}$.

**Control Task Configuration.**    To highlight the significance of the learned probing model, we follow the setting in Akyürek et al. (2022), and train the probing model with the linear function weights $\boldsymbol{w}$ fixed at 1. In this case, the probing model will only learn the statistics of the input $\boldsymbol{X}$, eliminating the need for in-context learning. Then during inference, we randomly sample $\boldsymbol{w} \sim \mathcal{N}(0, I_d)$ and record the resulting probing error. The minimal probing error for the control tasks are shown as dashed lines in Fig. 8. From the figure, the gap between the control task and the model probing results indicates the probing model utilizes the in-context information obtained in the output of each layer/loop iteration, suggesting that the probing error is significant.

## G    OPENML EXPERIMENT

To establish a connection with real-world applications and further validate our approach, we have conducted additional experiments using 10 datasets from OpenML (Vanschoren et al., 2013), which we believe better represent practical scenarios. The details of the used datasets are presented in Table 1.

Let $S = \{720, 725, 737, 803, 807, 816, 819, 847, 871, 42192\}$ be the set of all datasets. The datasets we examined have 0/1 binary labels. We trained the transformer and the looped transformer on 9 datasets and evaluated its in-context learning ability on the unseen test dataset. Both transformer and looped transformer have 256 embedding size, 8 heads. The tranformer has 12 layers, and looped transformer has 1 layer, trained to maximum loop iteration $b = 30$, and window size $T = 15$.

During training, we uniformly sampled prompts from 9 datasets, where for each prompt, we first randomly selected a training set, then randomly selected $k + 1$ samples from this training set, with $k$ being the number of in-context samples. During testing, we applied a similar approach for each test sample, selecting $k$ in-context samples from the test dataset, with care taken to exclude the test sample itself from these in-context pairs. The test accuracies are presented in Table 2.

| Dataset ID | Name | # Numerical Features | # Instances |
|---|---|---|---|
| 720 | abalone | 7 | 4177 |
| 725 | bank8FM | 8 | 8192 |
| 737 | space_ga | 8 | 3107 |
| 803 | delta_ailerons | 5 | 7129 |
| 807 | kin8nm | 8 | 8192 |
| 816 | puma8NH | 8 | 8192 |
| 819 | delta_elevators | 6 | 9517 |
| 847 | wind | 14 | 6574 |
| 871 | pollen | 5 | 3848 |
| 42192 | compas-two-years | 7 | 3848 |

Table 1: OpenML dataset used in the experiments. We only use numerical features for the input features.

As the result indicates, the looped transformer demonstrates comparable, and in some cases, better performance to the standard transformer in solving these OpenML datasets. We believe these experiments offer a more realistic representation of practical applications.

| Test Dataset ID | Train Dataset IDs | Transformer | Looped Transformer |
|---|---|---|---|
| 720 | $S\backslash\{720\}$ | $0.626 \pm 0.008$ | $\mathbf{0.662 \pm 0.008}$ |
| 725 | $S\backslash\{725\}$ | $0.511 \pm 0.007$ | $0.504 \pm 0.008$ |
| 737 | $S\backslash\{737\}$ | $0.656 \pm 0.006$ | $\mathbf{0.72 \pm 0.01}$ |
| 803 | $S\backslash\{803\}$ | $0.394 \pm 0.01$ | $0.40 \pm 0.01$ |
| 807 | $S\backslash\{807\}$ | $0.405 \pm 0.004$ | $\mathbf{0.416 \pm 0.005}$ |
| 816 | $S\backslash\{816\}$ | $0.463 \pm 0.004$ | $0.462 \pm 0.004$ |
| 819 | $S\backslash\{819\}$ | $0.483 \pm 0.005$ | $\mathbf{0.568 \pm 0.01}$ |
| 847 | $S\backslash\{847\}$ | $0.668 \pm 0.007$ | $\mathbf{0.757 \pm 0.006}$ |
| 871 | $S\backslash\{871\}$ | $\mathbf{0.532 \pm 0.004}$ | $0.51 \pm 0.005$ |
| 42192 | $S\backslash\{42192\}$ | $0.65 \pm 0.005$ | $0.65 \pm 0.008$ |

Table 2: The test accuracy for transformer and looped transformer, for different test dataset id. Looped transformer demonstrates comparable, and in some cases, better performance compared to vanilla transformer.

