# OpenReview forum: "Looped Transformers are Better at Learning Learning Algorithms"
_ICLR.cc/2024/Conference — ICLR 2024 poster_

### Official Review · Reviewer_QsDA · 2023-10-31

**Soundness:** 3 good
**Presentation:** 2 fair
**Contribution:** 3 good
**Rating:** 8
**Confidence:** 3

**Summary:**

In the paper the authors examine the applicability of looped transformers to the task of learning linear regression in context (giving sampled examples). They compare the performance of a looped transformer with a standard 12 layer transformer and a least squares solver. They show that for some settings the looped transformer matches or even outperforms the performance of a standard transformer on the task of linear regression, while incorporating significantly less parameters.

**Strengths:**

- It is an interesting approach to use looped transformers for the task of linear regression in-context. It seems to work well with significantly less parameters.
- The authors examine different settings, like using input injection, choosing the number of iterations, altering the number of layers, heads and the dimension of the embeddings.
- The experimental evaluation is convincing.

**Weaknesses:**

- The application of the looped transformer (in order to match the performance of a standard transformer) requires an extensive hyperparameter search for b and T. The authors make suggestions how this could be avoided, which should maybe be subject to further research.
- The structure of the paper can be improved, e.g., by ending the paper with a short conclusion instead of the related works.

Minor details:
- "It is worth noting" instead of "it's worth noting"
- "use the scheduling does not significantly impact the outcome" instead of "use the scheduling doesn’t significantly impact the outcome"
- The Figures are too small
- Fig. 2: The curve for the least squares solver is difficult to see
- Fig. 2 description: (left) and (right) should not be written after the punctuation in the sentence but before.

**Questions:**

- What is the experimental setting for the plots shown in Figure 2?
- Why are the curves for the looped transformer and the standard Transformer in Figure 2 exactly overlapping?
- How much time and resources does it take to find optimal values for b and T?
- If you consider the hyperparameter search is it still more useful to apply looped transformers instead of the standard transformers?

---

> ### Author Response · Authors · 2023-11-21
> **Response to Reviewer QsDA**
>
> Thank you for your feedback on our paper. We appreciate your comments and would like to address your concerns.
>
>
> > `an extensive hyperparameter search for b and T`: The authors make suggestions how this could be avoided, which should maybe be subject to further research.
>
> Thank you for pointing this out. An extensive hyperparameter search for $b$ and $T$ is subject to the goal of finding the appropriate values for $b$ and $T$ such that the looped transformer:
> - requires shortest training and inference time (related to the value of $b$),
> - occupies least training memory (related to the value of $T$),
> - is able to find a fixed-point solution beyond trained iterations.
>
>
> From our understanding of looped transformer training presented in the paper, the larger $b$ is, the better the looped transformer is at finding a fixed-point solution beyond trained iterations. Consequently, if the main objective of employing a looped transformer is to achieve this fixed-point solution, the strategy would involve progressively increasing $b$ until the solution is attained. This is akin to determining the optimal number of layers in a standard transformer.
>
> The search for $T$ mainly relates to balancing between achieving a fixed-point solution and avoiding training instability. Including regularization strategies can ease this instability, thus reducing the intensity of tuning for $T$.
>
> > How much time and resources does it take to find optimal values for b and T? If you consider the hyperparameter search is it still more useful to apply looped transformers instead of the standard transformers?
>
> As discussed in the above section, we believe if the goal is to find the value of $b$ and $T$ such that the looped TF can find a fixed-point solution, the burden of hyper-paremter search will be reduced by including the regularization strategies. So we would think there is no harm applying the looped transformer over the standard transformer.
>
> > `the structure of the paper can be improved, and minor details in the paper`: ending the paper with a short conclusion instead of the related works.
>
> Thanks for your suggestions. We have modified the draft accordingly.
>
> > What is the experimental setting for the plots shown in Figure 2? Why are the curves overlapping?
>
> In figure 2, we train the vanilla TF and looped TF on linear regression problem, with problem dimension $d=20$. In the left figure, we demonstrate the performance of looped transformer on in-context samples $k=40$. For the overlapping curve, we test the looped TF and standard TF on the same set of test prompts, therefore they have very similar performance.
>
> Thank you once again for your valuable feedback. We hope that our response effectively addresses your concerns.

---

> > ### Author Response · Authors · 2023-11-22
> >
> > Hi reviewer QsDA, we would like to inquire if our recent responses and revisions have adequately addressed your concerns regarding our ICLR submission. Could you please share your thoughts on the current rating of our paper? We value your feedback and appreciate your time and effort in this process.

---

### Official Review · Reviewer_xBYm · 2023-11-01

**Soundness:** 3 good
**Presentation:** 3 good
**Contribution:** 2 fair
**Rating:** 6
**Confidence:** 4

**Summary:**

This paper proposes a training methodology for looped transformers to effectively emulate iterative algorithms and provides empirical evidence that demonstrate the advantages of looped transformer on in-context learning. However, since all experiments are made on simulated datasets, whether the proposed method is effective in dealing with real-world data remains to be validated.

**Strengths:**

(1) The paper uses the looped transformer to emulate iterative learning algorithms and presents a novel methodology to train the looped transformer under reasonable assumptions.
(2) The paper provides a wide range of evaluation and detailed ablation studies of the proposed method on simulated datasets and demonstrates its superior performance compared to standard, non-recursive transformers.

**Weaknesses:**

(1) Since all experiments are made on simulated datasets, whether the proposed method is effective in dealing with real-world data remains to be validated.
(2) The classes of functions studied in the paper (including linear regression, decision tree, 2-layer ReLU NN, etc.) are ideal and relatively simple compared to the functions emerged in practical applications.

**Questions:**

1. Can you provide more details about the probability distribution over the used classes of functions, especially for decision trees and 2-layer ReLU NN?

---

> ### Author Response · Authors · 2023-11-21
> **Response to Reviewer xBYm (1/2)**
>
> We thank the reviewer for the thoughtful feedback and appreciate the opportunity to address your concerns.
>
> > `experiments are only on simluated datasets`: whether the proposed method is effective in dealing with real-world data remains to be validated. The classes of functions studied in the paper (including linear regression, decision tree, 2-layer ReLU NN, etc.) are ideal and relatively simple compared to the functions emerged in practical applications.
>
> Indeed, we acknowledge that the datasets in our study are comparatively simplistic. This is due to the focus of this paper is on training a transformer to solve data-fitting problem with simple function classes.
>
> To establish a connection with real-world applications and further validate our approach, we have conducted additional experiments using the following 10 datasets from [OpenML](https://www.openml.org/), which we believe better represent practical scenarios:
>
>
> | dataset id | Name | # numerical features | # instances |
> | -------- | -------- | -------- | -------- |
> | 720     | abalone     | 7     | 4177 |
> | 725     | bank8FM     | 8     | 8192 |
> | 737     | space_ga     | 8     | 3107 |
> | 803     | delta_ailerons     | 5     | 7129 |
> | 807     | kin8nm     | 8     | 8192 |
> | 816     | puma8NH     | 8     | 8192 |
> | 819     | delta_elevators     | 6     | 9517 |
> | 847     | wind     | 14     | 6574 |
> | 871     | pollen     | 5     | 3848 |
> | 42192     | compas-two-years     | 7     | 3848 |
>
> Let $S =$ {$720, 725, 737, 803, 807, 816, 819, 847, 871, 42192$} be the set of all datasets. The datasets we examined have 0/1 binary labels. We trained the transformer and the looped transformer on 9 datasets and evaluated its in-context learning ability on the unseen test dataset. Both transformer and looped transformer have 256 embedding size, 8 heads. The tranformer has 12 layers, and looped transformer has 1 layer, trained to maximum loop iteration $b=30$, and $T=15$.
>
> We uniformly sampled prompts from 9 datasets, where for each prompt, we first randomly selected a training set, then randomly selected $k+1$ samples from this training set, with $k$ being the number of in-context samples. During testing, we applied a similar approach for each test sample, selecting $k$ in-context samples from the test dataset, with care taken to exclude the test sample itself from these in-context pairs. The test accuracies are presented below:
>
> | test dataset id | train dataset ids | vanilla TF | looped TF |
> | -------- | -------- | -------- | -------- |
> | 720 | S \ {720} | 0.626 $\pm$ 0.008  | **0.662 $\pm$ 0.008** |
> | 725 | S \ {725} | 0.511 $\pm$ 0.007    | 0.504 $\pm$ 0.008 |
> | 737 | S \ {737} | 0.656 $\pm$ 0.006   | **0.72 $\pm$ 0.01** |
> | 803 | S \ {803} | 0.394 $\pm$ 0.01    | 0.40 $\pm$ 0.01 |
> | 807 | S \ {807} | 0.405 $\pm$ 0.004   | 0.416 $\pm$ 0.005 |
> | 816 | S \ {816} | 0.463 $\pm$ 0.004   | 0.462 $\pm$ 0.004 |
> | 819 | S \ {819} | 0.483 $\pm$ 0.005   | **0.568 $\pm$ 0.01** |
> | 847 | S \ {847} | 0.668 $\pm$ 0.007   | **0.757 $\pm$ 0.006**|
> | 871 | S \ {871} | **0.532 $\pm$ 0.004**   | 0.51 $\pm$ 0.005 |
> | 42192 | S \ {42192} | 0.65 $\pm$ 0.005  | 0.65 $\pm$ 0.008 |
>
> As the result indicates, the looped transformer demonstrates comparable, and in some cases, better performance to the standard transformer in solving these OpenML datasets. We have updated the draft with this experiment results, and we believe these experiments offer a more realistic representation of practical applications.
>
> In terms of further real-world applications, consider predicting user / customer behavior based on their historical data. Each customer represents a different context, and the optimal model form (linear, nonlinear, neural network, etc.) is often unknown. A transformer could learn an effective model for user data and then apply this learned model to new users. This is an area of ongoing investigation in our work, and we look forward to reporting our findings in the future.

---

> > ### Author Response · Authors · 2023-11-21
> > **Response to Reviewer xBYm (2/2)**
> >
> > > Can you provide more details about the probability distribution over the used classes of functions, especially for decision trees and 2-layer ReLU NN?
> >
> > We generate function classes following the approach outlined by [Garg et al.](https://arxiv.org/abs/2208.01066):
> >
> > 1. Linear function class: We sample $w \sim \mathcal{N}(0, I)$.
> > 2. Sparse linear function: We sample $w \sim \mathcal{N}(0, I)$, and randomly select $s=3$ coordinate to be non-zero.
> > 3. Decision tree: We utilize a binary decision tree with a depth of 4. For each non-leaf node, the decision to move left or right is based on the sign of one randomly selected coordinate of the input $x$. This coordinate is chosen uniformly at random. The leaf nodes are initialized following a normal distribution $\mathcal{N}(0, I)$. The evaluation of an input $x$ involves traversing from the root to a leaf node, moving left for a negative coordinate and right for a positive one, with the output being the value of the reached leaf node.
> > 4. 2-layer ReLU neural network: We initialize the weights of the network following $\mathcal{N}(0, I)$, and set the hidden layer dimension to 100.
> >
> > We have modified the draft to make it more clear as well.
> >
> > Thank you once again for your valuable feedback, and we hope that our response adequately addresses your concerns.

---

> > > ### Author Response · Authors · 2023-11-22
> > >
> > > Hi reviewer xBYm, we would like to inquire if our recent responses and revisions have adequately addressed your concerns regarding our ICLR submission. Could you please share your thoughts on the current rating of our paper? We value your feedback and appreciate your time and effort in this process.

---

### Official Review · Reviewer_gzpZ · 2023-11-03

**Soundness:** 2 fair
**Presentation:** 2 fair
**Contribution:** 2 fair
**Rating:** 5
**Confidence:** 4

**Summary:**

This paper proposes to use looped transformers to solve in-context learning, which achieves comparable performance to the standard transformer, but utilizes less than 10% of the parameters.

**Strengths:**

It is interesting to see looped transformers work well for in-context learning.

The paper provides a thorough evaluation and ablations of looped transformers for in-context learning.

The paper is well written.

**Weaknesses:**

My primary concern lies in the relevance of looped transformers to in-context learning. It appears that their main advantage is in reducing the number of parameters. However, it's not entirely clear why this reduction in parameters is crucial for in-context learning, especially in cases involving linear functions, sparse linear functions, random decision trees, and 2-layer ReLU networks. I find that the paper lacks in-depth mathematical insights or a thorough exploration of the practical implications that would help address this concern.

**Questions:**

What is the mathematical insight of looping transformers for in-context learning?

---

> ### Author Response · Authors · 2023-11-21
> **Response to Reviewer gzpZ**
>
> Thank you for your valuable feedback.
>
> > `relevance of looped transformers to in-context learning`: It’s not entirely clear why this reduction in parameters is crucial for in-context learning, especially in cases involving linear functions, sparse linear functions, random decision trees, and 2-layer ReLU networks
>
> We do not claim that a *reduction in parameters* is key to success in in-context learning. Instead, we aim to introduce a promising training method for a looped transformer, specifically tailored for in-context data fitting problems. The advantages of applying a looped transformer are twofold: 1) it reinforces the iterative nature commonly employed in solving data-fitting problems, and 2) it offers the benefit of saving parameters.
>
> > `lacks in-depth mathematical insights`: What is the mathematical insight of looping transformers for in-context learning?
>
> From the perspective of the expressive power of looped transformer: it has been extensively explored. [Bai et al., Theorem 3](https://arxiv.org/pdf/1909.01377.pdf) demonstrated that any traditional $L$-layer neural network can be represented by a weight-tied, input-injected network, highlighting the expressiveness of such networks. Furthermore, [Giannou et al.](https://arxiv.org/pdf/2301.13196.pdf) investigated how an encoder looped transformer can function as a universal computer. These studies provide solid theoretical foundations and constructions to demonstrate the significant expressiveness of transformers with looped architectures. Our work, instead, focuses on the empirical validation of this architecture, and presents a practical training method to effectively train a looped transformer architecture for in-context learning.
>
> From the perspective of applying a looped transformer to in-context data fitting problems: the most common and effective training algorithms are iterative and recursive in nature, such as gradient descent. Our insight is that equipping transformer architectures with a recursive (looped) structure enables them to learn policies for in-context learning that leverage the advantages of recursion.
>
> We hope our response address your concerns. We have updated the draft to include the discussion. Thanks again for your valuable insights and comments to our paper.

---

> ### Author Response · Authors · 2023-11-22
>
> Hi reviewer gzpZ, we would like to inquire if our recent responses and revisions have adequately addressed your concerns regarding our ICLR submission. Could you please share your thoughts on the current rating of our paper? We value your feedback and appreciate your time and effort in this process.

---

### Author Response · Authors · 2023-11-21
**General Response to AC and Reviewers**

We extend our gratitude to the reviewers for their constructive feedback and insightful inquiries. We are pleased that the reviewers recognized the interest in using looped transformers for in-context learning, as well as our comprehensive experimental analysis.

In response to the feedback, we have enriched the draft with an additional experiment focusing on a real-world application (OpenML dataset), restructured the paper, and made several other modifications for clarity. These changes are marked in blue in the updated draft for easy identification. We believe these updates adequately address your concerns and are grateful for the opportunity to contribute further to this field of research.

---

### Meta-Review · Area_Chair_kF1p · 2023-12-06

**Metareview:**

This work provides a training method for looped Transformers to perform in-context learning. The experimental results suggest that the looped Transformer can achieve performance comparable to standard Transformers in solving various tasks, mostly on simulation datasets. Most reviewers felt this work explores an interesting research direction, and the preliminary results are promising. Although there are still several concerns regarding the limited empirical evaluations, I felt the work is solid and vote for acceptance following the majority.

**Justification For Why Not Higher Score:**

The experimental results are limited and only conducted on simulation datasets.

**Justification For Why Not Lower Score:**

Most reviewers felt the work is interesting and the research direction is worth exploring.

---

### Decision · Program_Chairs · 2024-01-16

Accept (poster)